# CONVEXIFYING TRANSFORMERS: IMPROVING OPTIMIZATION AND UNDERSTANDING OF TRANSFORMER NETWORKS

## ABSTRACT

Understanding the fundamental mechanism behind the success of transformer networks is still an open problem in the deep learning literature. Although their remarkable performance has been mostly attributed to the self-attention mechanism, the literature still lacks a solid analysis of these networks and interpretation of the functions learned by them. To this end, we study the training problem of attention/transformer networks and introduce a novel convex analytic approach to improve the understanding and optimization of these networks. Particularly, we first introduce a convex alternative to the self-attention mechanism and reformulate the regularized training problem of transformer networks with our alternative convex attention. Then, we cast the reformulation as a convex optimization problem that is interpretable and easier to optimize. Moreover, as a byproduct of our convex analysis, we reveal an implicit regularization mechanism, which promotes sparsity across tokens. Therefore, we not only improve the optimization of attention/transformer networks but also provide a solid theoretical understanding of the functions learned by them. We also demonstrate the effectiveness of our theory through several numerical experiments.

## 1 INTRODUCTION

Transformer networks proposed by Vaswani et al. (2017) have become a dominant architecture in various tasks, especially Natural Language Processing (NLP) (Devlin et al., 2018; Radford et al., 2019), due to their extraordinary generalization properties and high capacity to learn from vast amount of data. Although there exists substantial empirical evidence on the effectiveness of transformer networks, revealing the underlying theoretical reasons behind their success is still an open research problem due to their highly nonlinear and nonconvex structure.

A significant body of research focused on analyzing certain components of transformer networks via empirical studies. As an example, Liu et al. (2021a); Vashishth et al. (2019); Dong et al. (2021); Voita et al. (2019); Takase et al. (2022); Liu et al. (2021a) studied the impact of the attention mechanism on transformer networks. Although these studies agreed that attention is an essential component of transformers, they also raised several issues regarding interpretability and optimization. Particularly, Voita et al. (2019) demonstrated that most attention heads can be removed without affecting the performance of the network, which is an indicator of large amount of redundancy in the network. Vashishth et al. (2019) provided a set of empirical evidence showing that attention might not be needed for some NLP tasks. Additionally, Dong et al. (2021) revealed that although attention is at the heart of transformer networks, training an attention network in the absence of Fully Connected Network (FCN) layers and skip connections is extremely challenging since the network output degenerates quickly without them. Similarly, Takase et al. (2022) discussed the importance of layer normalization and skip connections for transformer networks so that even changing the position of these might considerably impact the performance of a transformer network. However, a solid theoretical analysis of the underlying factors behind these issues is sill lacking, likely due to the highly complex and nonconvex structure of transformer networks.

A series of papers also focused on designing new alternatives to the self-attention mechanism which perform similarly and might provide further interpretations towards the overall model. One set

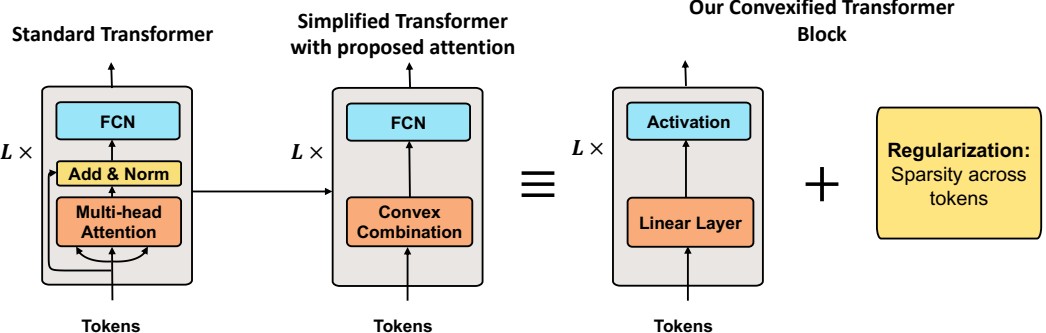

Figure 1: Summary of our main findings: We first propose an alternative to attention, i.e., taking the convex combinations of tokens, and then convexify whole transformer block (attention + Fully Connected Network (FCN)) with this new attention mechanism. The equivalent convex formulation also reveals a sparsity-inducing regularization across tokens as detailed in Theorem 1, 2, and 3.

of work utilizes multi-layer perceptron based architectures, (Tolstikhin et al., 2021; Tatsunami & Taki, 2021; Touvron et al., 2021; Liu et al., 2021b; Yu et al., 2021), while another set of of papers proposes Fourier based models (Lee-Thorp et al., 2021; Rao et al., 2021; Li et al., 2020; Guibas et al., 2021). Others also proposed replacing the self-attention mechanism with matrix decomposition (Geng et al., 2021). Although these works successfully applied to certain applications, they lack any solid theoretical analysis and understanding from an optimization perspective. Recently, Sahiner et al. (2022) attempted to analyzed transformer networks via convex duality by completely changing structure of the self-attention mechanism and removing FC layers. Even then, they failed to provide solid practical implications/benefits for transformers since their formulations are extremely challenging and complex to be trained in practice.

Recently, another line of research has focused on understanding structures and patterns emerge throughout the training of transformer networks (Power et al., 2022; Thilak et al., 2022; Barak et al., 2022). In particular, the grokking phenomenon was first observed by Power et al. (2022) on specific algorithmic tasks, such as modular division operations. Specifically, grokking refers to a sudden transition of validation or test accuracy to perfect generalization and this generalization happens well past the point of perfect training accuracy. This interesting behavior contradicts the common practice of early stopping in the training of deep learning models and definitely requires further understanding as to why this phenomenon emerges.

In order to remedy the issues associated with the standard transformer networks, in this paper, we develop a convex optimization perspective to train, analyze and understand transformer networks. Particularly, we first propose a convex alternative to the self-attention mechanism and then develop our convex analytic framework on the resulting model as detailed in Figure 1.

## 1.1 CONTRIBUTIONS

- We propose an alternative formulation to the standard self-attention mechanism and study the regularized training problem of attention/transformer networks with it.

- We convexify the regularized training problem of attention/transformer networks with the proposed attention layer as shown in Figure 1 and therefore enable globally optimal optimization without requiring nonconvex heuristics, e.g., normalization and skip connections.

- We also apply our convex analytic framework to various architectures, e.g., networks with or without an FCN layer. Thus, we are able to explain the impact of each component on the models learned throughout training.

- We reveal an implicit regularization mechanism induced by our attention mechanism. We further characterize this regularization as a sparsity-inducing factor across tokens.

- We demonstrate the effectiveness of our convex reformulation via various experimental results. We also show that our reformulation significantly mitigates the grokking phenomenon studied in recent papers (Power et al., 2022; Thilak et al., 2022).

## 1.2 NOTATIONS

We use lowercase and uppercase bold letters to denote vectors and matrices, respectively. We denote a certain column/element of a vector or matrix using subscripts. For example, $w_{jk}$ denotes the $jk^{th}$ entry of the matrix $\mathbf{W}$. We use $\mathbf{I}_k$ to denote the identity matrix of size $k \times k$ and $\mathbf{0}$ (or $\mathbf{1}$) to denote a vector/matrix of zeros (or ones) with appropriate sizes. We also use $[n]$ for the set of integers ranging from 1 to $n$. We represent the Euclidean and Frobenius norms as $\| \cdot \|_2$ and $\| \cdot \|_F$, respectively. We also use

Table 1: Notations.

| Notation | Description |
|----------|-------------|
| $N$ | # of sentences/samples |
| $n$ | # of tokens |
| $d$ | embedding dimension |
| $h$ | # of heads |
| $c$ | # of outputs |

$\mathbb{1}[x \geq 0]$ to denote the 0-1 valued indicator function. We also provide additional notations in Table 1.

## 2 TRANSFORMER NETWORKS

Given a data sample (or sentence) $\mathbf{X} \in \mathbb{R}^{h \times d}$ as a sequence of $h$ tokens with the embedding dimension $d$, we define the key, query, and value matrices as

$$\mathbf{Q} = \mathbf{X}\mathbf{W}_q, \quad \mathbf{W}_q \in \mathbb{R}^{d \times d}$$
$$\mathbf{K} = \mathbf{X}\mathbf{W}_k, \quad \mathbf{W}_k \in \mathbb{R}^{d \times d},$$
$$\mathbf{V} = \mathbf{X}\mathbf{W}_v, \quad \mathbf{W}_v \in \mathbb{R}^{d \times d}$$

which are the main components of the self-attention mechanism. Then, a single transformer block, which is basically a stack of self attention, residual connection, layer normalization, and point-wise feedforward connections, can be formulated as follows

$$
\begin{aligned}
\mathbf{A}_s &= \text{softmax}\left(\mathbf{Q}\mathbf{K}^\top\right)\mathbf{V} \\
\mathbf{A}_o &= \mathbf{A}_s\mathbf{W}_o, \quad \mathbf{W}_o \in \mathbb{R}^{d \times d} \\
\mathbf{X}_A &= \text{LayerNorm}\left(\mathbf{A}_o\right) + \mathbf{X} \\
\mathbf{X}_B &= \sigma\left(\mathbf{X}_A\mathbf{W}_1\right)\mathbf{W}_2
\end{aligned}
\quad , \tag{1}
$$

where $\sigma\left(\cdot\right)$ denotes the activation function for the FCN layer. Although skip connections, layer normalization and FCN also play a crucial role in a transformer block, the success of these networks has been mostly attributed to the self-attention part, denoted as $\mathbf{A}_o$ (Vaswani et al., 2017). Therefore, in the following section, we first study the training problem of a simplified transformer network, for which the network output is directly $\mathbf{A}_o$. We then extend our derivations to a transformer network with FCN layers.

## 3 ATTENTION-ONLY NETWORKS

We first consider a simplified transformer network only with a self attention layer that maps input sequence $\mathbf{X} \in \mathbb{R}^{n \times d}$ to the output sequence $\hat{\mathbf{Y}} \in \mathbb{R}^{n \times c}$ with $c$ outputs as

$$\hat{\mathbf{Y}} = \text{softmax}\left(\mathbf{X}\mathbf{W}_q\mathbf{W}_k^\top\mathbf{X}^\top\right)\mathbf{X}\mathbf{W}_v\mathbf{W}_o. \tag{2}$$

We call the model (2) as an attention-only network. This is a meaningful model and has been applied to various tasks, including machine translation, language modeling, image captioning, and object recognition (Vashishth et al., 2019).

We next consider a standard regression framework with an arbitrary convex loss function. Given a training set $\{\mathbf{X}_i, \mathbf{Y}_i\}_{i=1}^N$, where $\mathbf{X}_i \in \mathbb{R}^{n \times d}$ and $\mathbf{Y}_i \in \mathbb{R}^{n \times c}$ denote the input sequence and the labels/target outputs, respectively, the weight decay regularized training problem for the attention-only network in (2) is as follows

$$\min_{\{\mathbf{W}_\#\}} \sum_{i=1}^N \mathcal{L}\left(\text{softmax}\left(\mathbf{X}_i\mathbf{W}_q\mathbf{W}_k^\top\mathbf{X}_i^\top\right)\mathbf{X}\mathbf{W}_v\mathbf{W}_o, \mathbf{Y}_i\right) + \frac{\beta}{2}\sum_{\# \in \{q,k,v,o\}} \|\mathbf{W}_\#\|_F^2, \tag{3}$$

where $\mathcal{L}\left(\cdot\right)$ is an arbitrary convex loss function, including squared loss and cross entropy, and $\beta > 0$ is the regularization coefficient.

Although the attention-only model in (2) is quite powerful across various NLP tasks, e.g., natural language inference, neural machine translation, and text classification (Vashishth et al., 2019), the corresponding training problem in (3) is an extremely challenging optimization task and requires various nonconvex optimization heuristics to be adequately trained (Dong et al., 2021). To remedy these issues, in the following sections, we first reformulate the training problem by replacing the attention part with an alternative convex layer and then cast the reformulated training problem as a convex optimization problem that enables globally optimization of the network parameters.

## 3.1 CONVEX ATTENTION LAYER

We first note that since the $\mathrm{softmax}\,(\cdot)$ operation is highly nonlinear and nonconvex, the training problem in (3) is a challenging nonconvex optimization problem. Therefore, one may not adequately train attention networks. For example, Dong et al. (2021) shows that attention networks are likely to degenerate throughout the training and the output converges to a rank-1 matrix. Thus, they fail to learn the underlying tasks.

To avoid the issues associated with the nonconvex formulation in (2), we first replace the softmax operation with a simpler yet effective alternative. Particularly, since softmax converts the rows of its input matrix to a probability distribution, it can be relaxed as a linear operation with unit simplex constraints as follows

$$\text{for any } \mathbf{U} \in \mathbb{R}^{n \times n}, \quad \exists \mathbf{W} \in \Delta \text{ s.t. } \mathrm{softmax}\,(\mathbf{U})\,\mathbf{X} = \mathbf{W}\mathbf{X},$$

where $\Delta := \left\{\mathbf{W} \in \mathbb{R}^{n \times n} : \mathbf{w}_i \geq \mathbf{0}, \mathbf{1}^\top \mathbf{w}_i = 1, \forall i \in [n]\right\}$ denotes a convex set of constraints, also termed as unit simplex constraints. Thus, we simplified and convexified the attention mechanism without disturbing its structure. Therefore, (3) can be reformulated as follows

$$\min_{\substack{\mathbf{W}_1 \in \Delta \\ \mathbf{W}_2 \in \mathbb{R}^{d \times d}, \mathbf{W}_3 \in \mathbb{R}^{d \times c}}} \sum_{i=1}^{N} \mathcal{L}\left(\mathbf{W}_1 \mathbf{X}_i \mathbf{W}_2 \mathbf{W}_3, \mathbf{Y}_i\right) + \frac{\beta}{2}\left(\|\mathbf{W}_2\|_F^2 + \|\mathbf{W}_3\|_F^2\right). \tag{4}$$

Note that the model above utilizes a single head attention model and, therefore, may not be practically relevant due to its insufficient expressive power. Thus, we introduce the concept of head to the problem in (4) as follows

$$\min_{\substack{\mathbf{W}_{1j} \in \Delta \\ \mathbf{W}_{2j} \in \mathbb{R}^{d \times d}, \mathbf{W}_{3j} \in \mathbb{R}^{d \times c}}} \sum_{i=1}^{N} \mathcal{L}\left(\sum_{j=1}^{h} \mathbf{W}_{1j} \mathbf{X}_i \mathbf{W}_{2j} \mathbf{W}_{3j}, \mathbf{Y}_i\right) + \frac{\beta}{2}\left(\sum_{j=1}^{h} \|\mathbf{W}_{2j}\|_F^2 + \|\mathbf{W}_{3j}\|_F^2\right). \tag{5}$$

Now, we are ready to apply the convex analytic tools to (5) as detailed in the next section.

## 3.2 CONVEX OPTIMIZATION FOR ATTENTION-ONLY NETWORKS

As a warm-up, let us consider the scalar output prediction problem where the targets are one-dimensional, i.e., $y_i \in \mathbb{R}$. Then, (5) reduces to the following optimization problem

$$\min_{\substack{\mathbf{w}_{1j} \in \Delta \\ \mathbf{w}_{2j} \in \mathbb{R}^d, w_{3j} \in \mathbb{R}}} \sum_{i=1}^{N} \mathcal{L}\left(\sum_{j=1}^{h} \mathbf{w}_{1j}^\top \mathbf{X}_i \mathbf{w}_{2j} w_{3j}, y_i\right) + \frac{\beta}{2} \sum_{j=1}^{h}\left(\|\mathbf{w}_{2j}\|_2^2 + (w_{3j})^2\right). \tag{6}$$

Next, we first apply a rescaling between the parameters $\mathbf{w}_{2j}$ and $w_{3j}$ such that (6) can be described as an $\ell_1$ regularized optimization problem.

**Lemma 1.** *The problem in* (6) *is equivalent to the following $\ell_1$ regularized training problem*

$$\min_{\substack{\mathbf{w}_{1j} \in \Delta \\ \|\mathbf{w}_{2j}\|_2 \leq 1, w_{3j} \in \mathbb{R}}} \sum_{i=1}^{N} \mathcal{L}\left(\sum_{j=1}^{h} \mathbf{w}_{1j}^\top \mathbf{X}_i \mathbf{w}_{2j} w_{3j}, y_i\right) + \beta \|\mathbf{w}_3\|_1. \tag{7}$$

Using Lemma 1, the next theorem introduces a convex optimization problem that is equivalent to (6).

Table 2: Number of parameters and FLOPs for the convex and nonconvex models. Here, we use the following notations: $n$: # of tokens, $d$: embedding dimension, $h$: # of heads, and $c$: # of outputs.

| | Nonconvex | | | | Convex | |
|---|---|---|---|---|---|---|
| | Standard | | Alternative (Ours) | | | |
| | # of params | FLOPs | # of params | FLOPs | # of params | FLOPs |
| Scalar output | $h(3d^2 + d)$ | $\mathcal{O}(n^2dh)$ | $h(n+d+1)$ | $\mathcal{O}(nd)$ | $nd$ | $\mathcal{O}(nd)$ |
| Multi output | $h(3d^2 + dc)$ | $\mathcal{O}(n^2dh + ndhc)$ | $h(n+d+c)$ | $\mathcal{O}(nd+c)$ | $ndc$ | $\mathcal{O}(ndc)$ |
| Multi output with FCN | $h(3d^2 + dc)$ | $\mathcal{O}(n^2dh + ndhc)$ | $h(n+d+c)$ | $\mathcal{O}(nd+c)$ | $ndch$ | $\mathcal{O}(ndch)$ |

**Theorem 1.** *The nonconvex optimization problem* (6) *is equivalent to the following convex optimization problem*

$$\min_{\mathbf{Z} \in \mathbb{R}^{n \times d}} \frac{1}{2} \sum_{i=1}^{N} \mathcal{L}\left(\text{trace}\left(\mathbf{Z}^\top \mathbf{X}_i\right), y_i\right) + \beta \sum_{k=1}^{n} \|\mathbf{z}_k\|_2. \tag{8}$$

Note that the equivalent convex model in (8) requires a single parameter matrix $\mathbf{Z} \in \mathbb{R}^{n \times d}$, where each row is the attentions scores of the corresponding token. We also remark that the regularization in (8), i.e., the sum of $\ell_2$ norms of the rows of the parameter matrix $\mathbf{Z}$, is a specific type of regularization, also known as group $\ell_1$ or Lasso, introduced by Bakin et al. (1999) and shown to promote group sparsity across parameters (Yuan & Lin, 2006). In our case, the group sparsity is across the token index $k$. Therefore, one can interpret the model in (8) as a sparse linear model, where the sparsity is across tokens. In other words, (8) can be explained as a model that tries to use as few tokens as possible to fit the training labels $\{y_i\}_{i=1}^{N}$.

Unlike the nonnegative attention scores in (6), denoted as $\mathbf{w}_{1j} \in \Delta$, the convex parameters $\mathbf{Z} \in \mathbb{R}^{n \times d}$ do not require any constraints. Therefore, one can directly apply standard training algorithms, such as SGD and Adam, to train the convex problem (8). Moreover, an optimal set of parameters for (6) can be recovered from a solution to (8) as proven in the following result.

**Proposition 1.** *After solving the convex optimization problem in* (8)*, one can recover an optimal solution to the nonconvex optimization problem in* (6)*, denoted as* $\{\mathbf{w}_{1j}^*, \mathbf{w}_{2j}^*, w_{3j}^*\}_{j=1}^{h}$*, as follows*

$$\mathbf{w}_{1j}^* = \mathbf{e}_j, \ \mathbf{w}_{2j}^* = \frac{\mathbf{z}_j}{\sqrt{\|\mathbf{z}_j\|_2}}, \ w_{3j}^* = \sqrt{\|\mathbf{z}_j\|_2}, \forall j \in [h],$$

*where* $\mathbf{e}_j \in \mathbb{R}^n$ *is the* $j^{th}$ *ordinary basis vector,* $\mathbf{z}_j \in \mathbb{R}^d$ *is the* $j^{th}$ *row of* $\mathbf{Z}$*, and we assume that there are* $h$ *nonzero rows out of* $n$ *rows of* $\mathbf{Z}$ *due to the sparsity-inducing regularization in* (8)*.*

Proposition 1 proves that there is a one-to-one mapping between the parameters of the nonconvex formulation in (6) and the convex formulation in (8). Therefore, there is no need to solve the challenging nonconvex optimization problem (6) which also requires several optimization heuristics to be adequately trained. Instead, one can solve the convex problem (8) and then use the mapping in Proposition 1 to obtain an optimal solution to (6).

### 3.3 EXTENSION TO MULTIDIMENSIONAL OUTPUTS

In the previous section, we considered a setting with scalar target variables, i.e., $y_i \in \mathbb{R}$. However, for some problems, e.g., multiclass classification, target variables can be multidimensional. Therefore, we now extend the analysis to the problems with multiple/vector outputs as follows

$$\min_{\substack{\mathbf{w}_{1j} \in \Delta \\ \mathbf{w}_{2j} \in \mathbb{R}^d, \mathbf{w}_{3j} \in \mathbb{R}^c}} \sum_{i=1}^{N} \mathcal{L}\left(\sum_{j=1}^{h} \mathbf{w}_{1j}^\top \mathbf{X}_i \mathbf{w}_{2j} \mathbf{w}_{3j}, \mathbf{y}_i\right) + \frac{\beta}{2} \sum_{j=1}^{h} \left(\|\mathbf{w}_{2j}\|_2^2 + \|\mathbf{w}_{3j}\|_1^2\right), \tag{9}$$

where $\mathbf{y}_i \in \mathbb{R}^c$ and $c$ denotes the number of outputs/classes. Note that here we put $\ell_1^2$-norm on $\mathbf{w}_{3j}$ to enable our convex arguments but this does not impact performance of the network in practice. Then, following the same derivations yields the convex program in the next result.

**Theorem 2.** *The nonconvex optimization problem* (9) *is equivalent to the following convex optimization problem*

$$\min_{\mathbf{Z}_l \in \mathbb{R}^{n \times d}} \sum_{i=1}^{N} \sum_{l=1}^{c} \mathcal{L}\left(\text{trace}\left(\mathbf{Z}_l^\top \mathbf{X}_i\right), y_{il}\right) + \beta \sum_{l=1}^{c} \sum_{k=1}^{n} \|\mathbf{z}_{lk}\|_2. \tag{10}$$

Theorem 2 shows that the equivalent convex model becomes separable over the output index $l$, i.e., instead of a single parameter matrix in (8), here we have $c$ parameter matrices due to having $c$ outputs in the nonconvex model (9) (see Table 2 for details). This also illustrates that the number of outputs in the network directly controls the overparameterization level of the convex formulation.

### 3.4 ATTENTION NETWORKS WITH FCN LAYERS

Although the model in (5) exhibits interesting properties in various applications (Dong et al., 2021), it is basically a linear function of the token matrix $\mathbf{X}$. Therefore, it is likely to suffer from inadequate performance especially for some challenging problems in NLP. A series of papers (Dong et al., 2021; Geva et al., 2021; Meng et al., 2022; Geva et al., 2022b;a) also confirmed the importance of FCNs via extensive empirical evidence. Therefore, in this section, we include FCN layers to our attention only model in (5) and derive an equivalent convex formulation for this model.

Here, we consider the following optimization problem

$$\min_{\substack{\mathbf{w}_{1j} \in \Delta \\ \mathbf{w}_{2j}, \mathbf{w}_{3j} \in \mathbb{R}^c}} \sum_{i=1}^{N} \mathcal{L}\left(\sigma\left(\sum_{j=1}^{h} \mathbf{w}_{1j}^\top \mathbf{X}_i \mathbf{w}_{2j}\right)\mathbf{w}_{3j}, \mathbf{y}_i\right) + \frac{\beta}{2}\left(\sum_{j=1}^{h} \|\mathbf{w}_{2j}\|_2^2 + \|\mathbf{w}_{3j}\|_1^2\right), \tag{11}$$

where $\sigma\left(\cdot\right)$ is the activation function.

**Theorem 3.** *The nonconvex optimization problem* (11) *with the gated ReLU activation is equivalent the following convex optimization problem*

$$\min_{\mathbf{Z}_{jl} \in \mathbb{R}^{n \times d}} \sum_{i=1}^{N} \sum_{l=1}^{c} \mathcal{L}\left(\sum_{j=1}^{h} \mathbb{1}_{ij}\text{trace}\left(\mathbf{Z}_{jl}^\top \mathbf{X}_i\right), y_{il}\right) + \beta \sum_{l=1}^{c} \sum_{j=1}^{h} \sum_{k=1}^{n} \|\mathbf{z}_{jlk}\|_2, \tag{12}$$

*where* $\mathbb{1}_{ij} := \mathbb{1}\left\{\mathbf{u}_{1j}^\top \mathbf{X}_i \mathbf{u}_{2j} \geq 0\right\}$ *denotes the indicator function for gated ReLU activation and here* $\left\{\mathbf{u}_{1j}, \mathbf{u}_{2j}\right\}_{j=1}^{h}$ *are fixed vectors that can be randomly selected.*

Theorem 3 implies that introducing the activation function further increases in the overparameterization level of the equivalent convex formulation. Precisely, (12) has $h$ times more parameters than (10) as shown in Table 2.

## 4 NUMERICAL EXPERIMENTS

In this section, we present experimental results corroborating our theory in the previous sections.

**Student-teacher setting with BERT:** We first consider a student-teacher setting with the pretrained BERT model in the Hugging Face repository, i.e., `bert-base-uncased`. Particularly, we feed the samples from the mrpc subset of the glue dataset (Warstadt et al., 2018; Wang et al., 2019) through the pretreained BERT model and save the input and output activations in a certain layer. Then, we train the attention-only models, i.e., standard nonconvex self-attention (3), alternative nonconvex attention (9), and convex (10), from scratch using these pre and post activations as our training dataset. All the experiments throughout this section are performed using a single GPU on Google Colab. We also use the same regularization coefficient $\beta$ and optimizer, i.e., Adam, and tune the learning rate and regularization coefficient by performing a grid search on a validation dataset for both algorithms. However, notice that we do not use any nonconvex optimization heuristics, e.g., layer normalization and skip connections, for the convex model in all the experiments. In Figure 2, we plot the objective values (i.e. training loss + regularization term) and test losses with respect to time in seconds using the data extracted from the sixth layer of pretrained BERT model. We observe

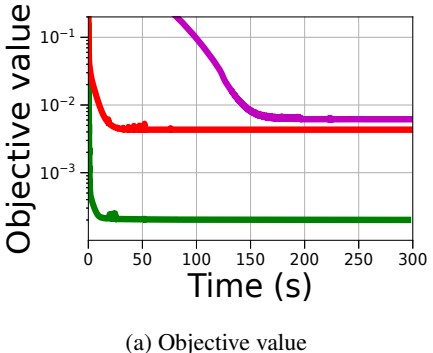
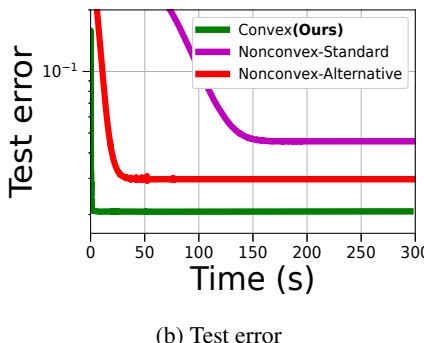

(a) Objective value

(b) Test error

Figure 2: Comparison of the convex and nonconvex models on the dataset extracted from pretrained BERT architecture in a student-teacher setting. Here, include two non-convex models, specifically standard self-attention model (Nonconvex-Standard) in (3) and alternative attention (Nonconvex-Alternative) in (9). Our convex training approach achieves significantly lower objective value and test error than the original nonconvex training.

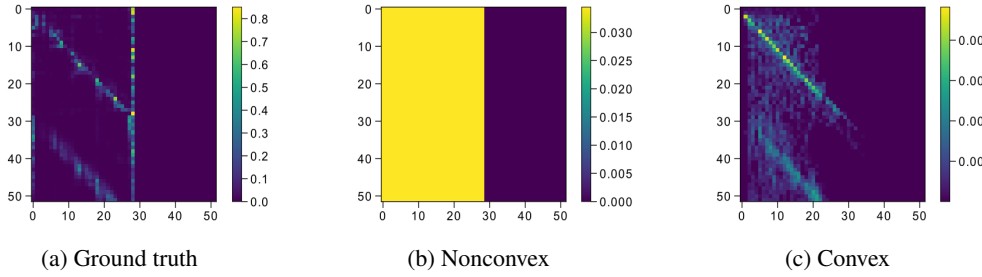

(a) Ground truth

(b) Nonconvex

(c) Convex

Figure 3: Attention maps obtained by our convex and standard nonconvex training approaches as well as the ground truth attention map in the BERT model. Here, the nonconvex training fails to learn the underlying patterns and simply achieves a uniform attention map. However, our convex training approach outputs an attention map that is close to the ground truth.

that our convex training approach achieves almost an order of magnitude smaller objective value than the standard nonconvex training, which is possibly stuck at a local minimum. This effectiveness in training also translates into better generalization, i.e., our convex training approach obtains a lower test loss than the standard nonconvex training. In order to understand the functions learned by each models, we also analyzed the attention maps in Figure 3. Here, standard nonconvex training fails to learn the underlying model and outputs a uniform attention map across token. However, our convex training outputs an attention map that is quite similar to the ground truth attention map, and therefore we successfully learn the structure in the training data. Hence, these experiments clearly illustrate the effectiveness of our convex training approach in both training and testing.

**Algorithmic datasets and Grokking:** Inspired by the grokking phenomenon observed in Power et al. (2022), we next validate the effectiveness of our convex training approach against standard transformer networks with the self-attention mechanism in (1) on algorithmic datasets. Particularly, we use the same setting in Power et al. (2022), and evaluate the performance on modular division operations with $\mod 97$ and $\mod 15$, where we train the architectures till they reach $99\%$ test accuracy whenever possible. In Figure 4, we first replicate the results in Power et al. (2022) and confirm that the grokking phenomenon indeed emerges here, i.e., the nonconvex curve (purple) reaches $100\%$ training accuracy at around $10^3$ iterations in Figure 4a while it requires more than $10^5$ iterations to reach perfect generalization in Figure 4b. We also compare the nonconvex and convex training approaches. Here, we show that our convex training approach converges to the perfect generalization accuracy $10\times$ faster than the nonconvex one in Figure 4b. Moreover, the convex model also yield significantly lower test loss in Figure 4c, which implies that it has higher confidence in test predictions and therefore more robust than standard nonconvex training.

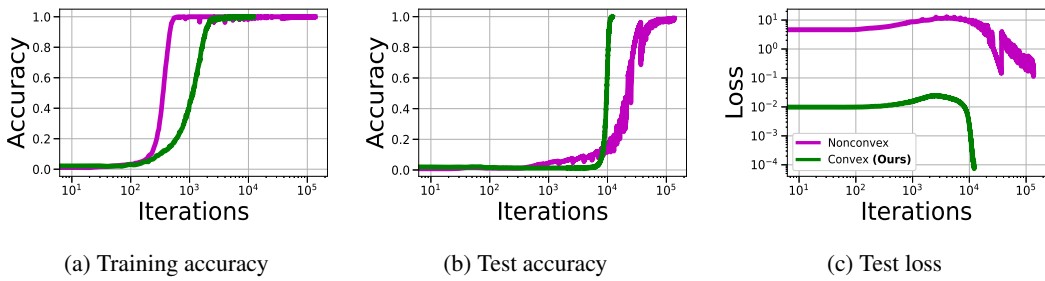

(a) Training accuracy          (b) Test accuracy          (c) Test loss

Figure 4: Comparison of the convex and nonconvex models on the modular division operation mod 97. Here, we train the networks to reach 99% test accuracy and show that our convex training approach exhibits a significantly faster convergence and lower test loss than the nonconvex training.

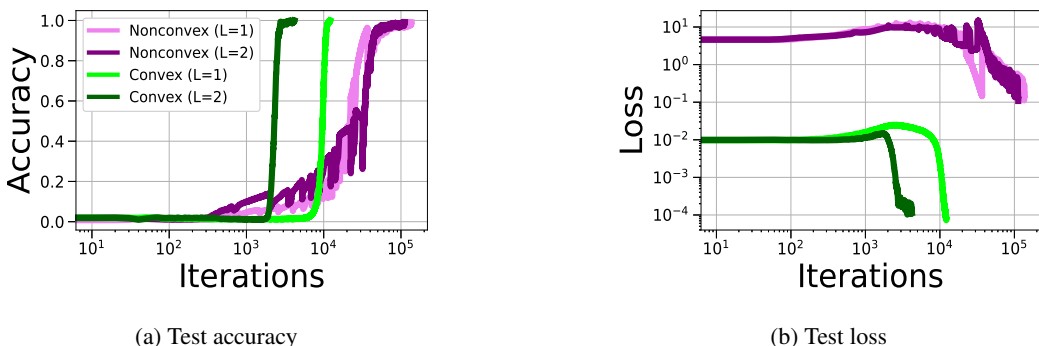

(a) Test accuracy          (b) Test loss

Figure 5: Comparison of one- and two-layer transformer networks on the modular division task mod 97, where $L$ denotes the number of layers in each model. We observe that introducing one more layer substantially improves the convergence speed of our convex formulation while it fails to make a noticeable impact on the nonconvex formulation.

We remark that in the previous section, we theoretically analyze only single attention/transformer blocks. However, since the benign impact of depth or number of layers (denoted as $L$) has already been empirically proven in the deep learning literature, we also propose an extension of our convex model to deeper settings. We basically stack the convex transformer layers in (12) to obtain an arbitrarily deep network. In Figure 5, we compare the performance of two-layer transformer networks with one-layer networks. Here, we observe that while adding one more layer results in significant improvements for the convex model, especially in terms of optimization speed, it fails to make any discernible difference for the nonconvex model. Moreover, we run the algorithms on the mod 15 operation which is basically more challenging task due to smaller number of samples. In this case, one-layer models are not able learn the underlying task perfectly as demonstrated in Figure 6 but our convex model is significantly better in terms of both test accuracy and test loss. By increasing the number of layers to four, we enable both models to achieve perfect generalization accuracy. Our deep model is much faster and also yields lower test loss than the nonconvex model.

We next empirically analyze the grokking phenomenon on both our convex and standard nonconvex models. For this purpose, we plot the number of iterations to reach 99% test accuracy for each of our experiments in Figure 7a. Notice that here we do not include the one-layer results for the mod 15 case, since both models fail to achieve perfect generalization in that case. Figure 7a clearly shows that our convex training approach converges to the 99% accuracy level substantially faster than the standard nonconvex training. Therefore, we also mitigate the impact of the grokking phenomenon as demonstrated in Figure 7b, where we quantify the amount of grokking in terms of the number of iterations. Thus, we also conjecture that the grokking phenomenon can be mostly attributed to the nonlinear and nonconvex structure of standard transformer models.

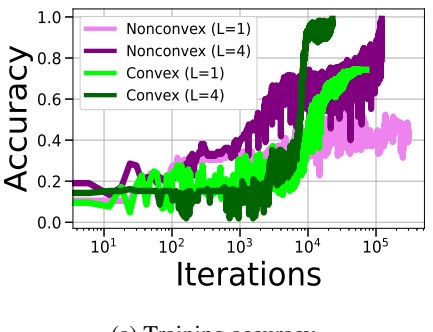

(a) Training accuracy

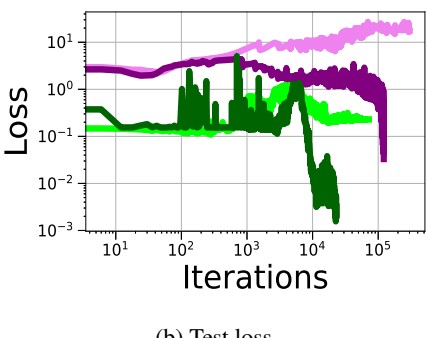

(b) Test loss

Figure 6: Comparison of one- and four-layer transformer networks on the modular division task $\mod 15$. Here, one-layer networks fail to achieve $99\%$ test accuracy however our convex training approach (light green) still generalizes better than the nonconvex training (light purple). We also show that perfect generalization accuracy can be achieved with four layers.

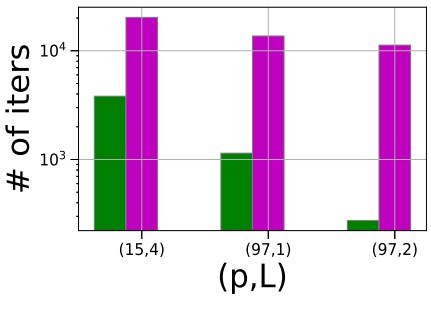

(a) Training iterations

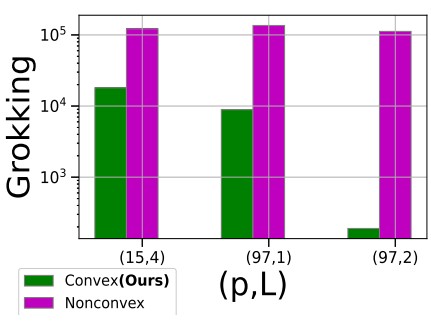

(b) Grokking iterations

Figure 7: Amount of grokking in terms of # of iterations required by our convex and standard nonconvex training approaches, where $p$ and $L$ denote the coefficient for the modular division and the number of layers, respectively. Here, we do not include the one-layer results in Figure 6 since both algorithms fail to achieve $99\%$ test accuracy. We demonstrate that the impact of grokking is substantially mitigated with our convex training approach.

## 5 CONCLUSION

In this paper, we studied the regularized training problem of attention/transformer networks and developed a convex analytic framework to train these networks. Particularly, we first proposed a convex alternative to the self-attention mechanism and then reformulated the training problem with this alternative attention mechanism as convex optimization problems. Thanks to our convex reformulation, we globally optimize the network parameters without requiring any kind of nonconvex optimization heuristics. In addition, the functions learned by our reformulation is transparent and interpretable. More importantly, the reformulated problem reveals a sparsity-inducing regularization mechanism across tokens in the data, which also sheds more light on the structure of the resulting function and its generalization properties. We then empirically verified effectiveness of our convex training approach over standard nonconvex training via several numerical experiments.

We also note that analyzing transformer networks through the lens of convex optimization theory is extremely crucial since it may result in substantial improvements in the understanding and optimization of these networks. However, it is also quite challenging due to the inherent nonconvex structure of the network model. To the best of our knowledge, this paper is the first step in this direction and therefore has some limitations which can hopefully be eliminated by future work. Specifically, in this paper, we mainly focused on the theory side of convex analysis and empirically validated the theory on a few small-scale problem instances. We hope that a comprehensive and large-scale empirical verification of our theory will be conducted by the follow-up papers.

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

# Appendix

## Table of Contents

## A   PROOFS OF THE RESULTS IN THE MAIN PAPER

### A.1   PROOF OF LEMMA 1

We first note that similar scaling techniques were previously studied in several papers, e.g., Lemma 1 of Ergen & Pilanci (2021), Theorem 1 of Neyshabur et al. (2014), Section 2 of Savarese et al. (2019), equation (2-3) of Pilanci & Ergen (2020).

We start with restating the optimization problem as follows

$$\min_{\substack{\mathbf{w}_{1j} \in \Delta \\ \mathbf{w}_{2j} \in \mathbb{R}^d, w_{3j} \in \mathbb{R}}} \sum_{i=1}^N \mathcal{L}\left( \sum_{j=1}^h \mathbf{w}_{1j}^\top \mathbf{X}_i \mathbf{w}_{2j} w_{3j}, y_i \right) + \frac{\beta}{2} \sum_{j=1}^h \left( \|\mathbf{w}_{2j}\|_2^2 + (w_{3j})^2 \right), \qquad (13)$$

We first apply the following scaling for $\{\mathbf{w}_{2j}, w_{3j}\}_{j=1}^m$

$$\bar{\mathbf{w}}_{2j} := \alpha_j \mathbf{w}_{2j}, \ \bar{w}_{3j} := \frac{w_{3j}}{\alpha_j}. \qquad (14)$$

where $\alpha_j > 0$. Since this scaling doesn't change the output of the network, i.e.,

$$\sum_{j=1}^h \mathbf{w}_{1j}^\top \mathbf{X}_i \bar{\mathbf{w}}_{2j} \bar{w}_{3j} = \sum_{j=1}^h \mathbf{w}_{1j}^\top \mathbf{X}_i \mathbf{w}_{2j} w_{3j},$$

the training loss part of the objective function stays the same. Thus, we can directly search for the optimal scaling parameter $\alpha_j > 0$ by minimizing the regularization term via the following AM-GM inequality

$$\sum_{j=1}^h \left( \|\bar{\mathbf{w}}_{2j}\|_2^2 + (\bar{w}_{3j})^2 \right) = \sum_{j=1}^h \left( \alpha_j^2 \|\mathbf{w}_{2j}\|_2^2 + \frac{(w_{3j})^2}{\alpha_j^2} \right) \geq 2 \sum_{j=1}^h \left( \|\mathbf{w}_{2j}\|_2 |w_{3j}| \right) = 2 \sum_{j=1}^h \left( \|\bar{\mathbf{w}}_{2j}\|_2 |\bar{w}_{3j}| \right)$$

where the equality is achieved when $\alpha_j = \sqrt{\frac{|w_{3j}|}{\|\mathbf{w}_{2j}\|_2}}$. Thus, we obtain a reformulation of (13) where the regularization term is in a multiplicative form as follows

$$\min_{\substack{\mathbf{w}_{1j} \in \Delta \\ \mathbf{w}_{2j} \in \mathbb{R}^d, w_{3j} \in \mathbb{R}}} \sum_{i=1}^N \mathcal{L}\left( \sum_{j=1}^h \mathbf{w}_{1j}^\top \mathbf{X}_i \mathbf{w}_{2j} w_{3j}, y_i \right) + \beta \sum_{j=1}^h \|\mathbf{w}_{2j}\|_2 |w_{3j}|. \qquad (15)$$

Next, we apply a variable change to the reformulation in (15) as follows

$$\mathbf{w}'_{2j} := \frac{\mathbf{w}_{2j}}{\|\mathbf{w}_{2j}\|_2}, \quad w'_{3j} := w_{3j} \|\mathbf{w}_{2j}\|_2.$$

With this variable change, we rewrite (15) as

$$\min_{\substack{\mathbf{w}_{1j}\in\Delta \\ \mathbf{w}'_{2j}:\left\|\mathbf{w}'_{2j}\right\|_2=1 \\ w'_{3j}\in\mathbb{R}}} \sum_{i=1}^{N}\mathcal{L}\left(\sum_{j=1}^{h}\mathbf{w}_{1j}^{\top}\mathbf{X}_i\mathbf{w}'_{2j}w'_{3j},y_i\right)+\beta\sum_{j=1}^{h}|w'_{3j}|. \tag{16}$$

This concludes the proof and yields the following equivalent formulation of (16)

$$\min_{\substack{\mathbf{w}_{1j}\in\Delta \\ \mathbf{w}'_{2j}:\left\|\mathbf{w}'_{2j}\right\|_2=1 \\ w'_{3j}\in\mathbb{R}}} \sum_{i=1}^{N}\mathcal{L}\left(\sum_{j=1}^{h}\mathbf{w}_{1j}^{\top}\mathbf{X}_i\mathbf{w}'_{2j}w'_{3j},y_i\right)+\beta\left\|\mathbf{w}'_3\right\|_1.$$

We also note that the equality constraint $\left\|\mathbf{w}'_{2j}\right\|_2=1$ can be relaxed as $\left\|\mathbf{w}'_{2j}\right\|_2\leq 1$ due to the optimality conditions arising from the regularization term $\left\|\mathbf{w}'_3\right\|_1$. $\qquad\square$

### A.2 PROOF OF PROPOSITION 1

We first note that in order to maintain strong duality in our convex problem derivations, we basically use the arguments in Section A.4, where the we prove that as long as $h$ exceeds a certain threshold $h^*$, there will be sparsity in the solution due to the sparsity-inducing regularization in (8). And we have the following upperbound $h^*\leq N+1$. Note that this $N+1$ upperbound is the worst case scenario and $h^*\ll N+1$ in practice as validated in Pilanci & Ergen (2020). Thus, below, we assume that there is a sparsity pattern in the solution.

Given an optimal solution to (8), denoted as $\mathbf{Z}^*\in\mathbb{R}^{n\times d}$, we first rewrite this solution as a summation of rank-1 matrices as follows

$$\mathbf{Z}^*=\sum_{j=1}^{h}\mathbf{e}_j\mathbf{z}_j^{\top}=\sum_{j=1}^{h}\mathbf{e}_j\frac{\mathbf{z}_j^{\top}}{\sqrt{\left\|\mathbf{z}_j\right\|_2}}\sqrt{\left\|\mathbf{z}_j\right\|_2}$$

where $\mathbf{e}_j\in\mathbb{R}^n$ is the $j^{th}$ ordinary basis vector and we assume that there are $h$ nonzero rows out of $n$ rows of $\mathbf{Z}$ due to the sparsity-inducing regularization in (8). Then, this implies that the output of the optimal can be equivalently formulated as follows

$$\begin{aligned}\text{trace}\left(\mathbf{Z}^{*\top}\mathbf{X}_i\right)&=\sum_{j=1}^{h}\mathbf{e}_j^{\top}\mathbf{X}_i\frac{\mathbf{z}_j}{\sqrt{\left\|\mathbf{z}_j\right\|_2}}\sqrt{\left\|\mathbf{z}_j\right\|_2}\\&=\sum_{j=1}^{h}\mathbf{w}_{1j}^{*\top}\mathbf{X}_i\mathbf{w}_{2j}^*w_{3j}^*\end{aligned}\quad\Longrightarrow\quad \mathbf{w}_{1j}^*=\mathbf{e}_j,\ \mathbf{w}_{2j}^*=\frac{\mathbf{z}_j}{\sqrt{\left\|\mathbf{z}_j\right\|_2}},\ w_{3j}^*=\sqrt{\left\|\mathbf{z}_j\right\|_2},$$

where $\{\mathbf{w}_{1j}^*,\mathbf{w}_{2j}^*,w_{3j}^*\}_{j=1}^{h}$ denotes an optimal solution to (6).

Next, we show that both of these solution sets achieve the same objective value

$$\begin{aligned}f\left(\{\mathbf{X}_i,y_i\}_{i=1}^{N}\right)&:=\sum_{i=1}^{N}\mathcal{L}\left(\sum_{j=1}^{h}\mathbf{w}_{1j}^{*\top}\mathbf{X}_i\mathbf{w}_{2j}^*w_{3j}^*,y_i\right)+\frac{\beta}{2}\sum_{j=1}^{h}\left(\left\|\mathbf{w}_{2j}^*\right\|_2^2+\left(w_{3j}^*\right)^2\right)\\&=\sum_{i=1}^{N}\mathcal{L}\left(\sum_{j=1}^{h}\mathbf{e}_j^{\top}\mathbf{X}_i\frac{\mathbf{z}_j}{\sqrt{\left\|\mathbf{z}_j\right\|_2}}\sqrt{\left\|\mathbf{z}_j\right\|_2},y_i\right)+\frac{\beta}{2}\sum_{j=1}^{h}\left(\left\|\frac{\mathbf{z}_j}{\sqrt{\left\|\mathbf{z}_j\right\|_2}}\right\|_2^2+\left(\sqrt{\left\|\mathbf{z}_j\right\|_2}\right)^2\right)\\&=\sum_{i=1}^{N}\mathcal{L}\left(\sum_{j=1}^{h}\mathbf{e}_j^{\top}\mathbf{X}_i\mathbf{z}_j,y_i\right)+\frac{\beta}{2}\sum_{j=1}^{h}\left(\left\|\mathbf{z}_j\right\|_2+\left\|\mathbf{z}_j\right\|_2\right)\\&=\sum_{i=1}^{N}\mathcal{L}\left(\sum_{j=1}^{h}\text{trace}\left(\mathbf{z}_j\mathbf{e}_j^{\top}\mathbf{X}_i\right),y_i\right)+\beta\sum_{j=1}^{h}\left\|\mathbf{z}_j\right\|_2\\&=\sum_{i=1}^{N}\mathcal{L}\left(\text{trace}\left(\mathbf{Z}^{*\top}\mathbf{X}_i\right),y_i\right)+\beta\sum_{k=1}^{n}\left\|\mathbf{z}_k\right\|_2,\end{aligned}\tag{17}$$

where the last inequality follows from the fact that there are $h$ nonzero rows out of $n$ rows of $\mathbf{Z}$ due to the sparsity-inducing regularization in (8). Note that (17) and (8) are the same objectives evaluated at $\mathbf{Z}^*$, which concludes the proof.

**Extension to multidimensional outputs in Section 3.3:** Here we show that the proof above can be straightforwardly extended to the multidimensional output case in Section 3.3.

Given an optimal solution to (10), denoted as $\mathbf{Z}_l^* \in \mathbb{R}^{n \times d}$, we first rewrite this solution as a summation of rank-1 matrices as follows

$$\mathbf{Z}_l^* = \sum_{k=1}^{h} \mathbf{e}_{lk}\mathbf{z}_{lk}^\top = \sum_{k=1}^{h} \mathbf{e}_{lk} \frac{\mathbf{z}_{lk}^\top}{\sqrt{\|\mathbf{z}_{lk}\|_2}} \sqrt{\|\mathbf{z}_{lk}\|_2}.$$

Then, this implies that the output of the optimal can be equivalently formulated as follows

$$\text{trace}\left(\mathbf{Z}_l^{*\top}\mathbf{X}_i\right) = \sum_{k=1}^{h} \mathbf{e}_{lk}^\top \mathbf{X}_i \frac{\mathbf{z}_{lk}}{\sqrt{\|\mathbf{z}_{lk}\|_2}} \sqrt{\|\mathbf{z}_{lk}\|_2} \implies \mathbf{w}_{1j}^* = \mathbf{e}_{lk}, \; \mathbf{w}_{2j}^* = \frac{\mathbf{z}_{lk}}{\sqrt{\|\mathbf{z}_{lk}\|_2}}, \; \mathbf{w}_{3j}^* = \mathbf{e}_l \sqrt{\|\mathbf{z}_{lk}\|_2},$$

where $\{\mathbf{w}_{1j}^*, \mathbf{w}_{2j}^*, \mathbf{w}_{3j}^*\}_{j=1}^{hc}$ denotes an optimal solution to (9). Note that here index $j \in [hc]$ instead of $j \in [h]$ in the scalar output case. $\qquad\square$

## A.3 Proof of Theorem 1

We first provide a summary of our proof strategy. For the derivations of the convex formulation, we basically need to find the bidual form of (6), i.e., the dual of the dual problem. Thus, we start with taking the dual of (6). To avoid nonconvexity in the dual problem, we reformulate the dual constraint, which makes the problem nonconvex, as a convex constraint. Therefore, we obtain a convex dual problem. Then, we take the dual of the dual problem to get the bidual formulation of (6). Since we convexify the dual problem, the bidual formulation is also a convex problem. Therefore, we achieve an equivalent convex formulation of the original nonconvex training problem (6). We also note that a similar proof strategy was also used in Pilanci & Ergen (2020).

In order to take the dual of (7) (i.e. restated below for the convenience of the reader) we need to form the Lagrangian function for the following optimization problem

$$\min_{\substack{\mathbf{w}_{1j} \in \Delta \\ \|\mathbf{w}_{2j}\|_2 \leq 1, w_{3j} \in \mathbb{R}}} \sum_{i=1}^{N} \mathcal{L}\left(\sum_{j=1}^{h} \mathbf{w}_{1j}^\top \mathbf{X}_i \mathbf{w}_{2j} w_{3j}, y_i\right) + \beta \|\mathbf{w}_3\|_1.$$

To construct the Lagrangian function, we first introduce an additional variable $\hat{\mathbf{y}} \in \mathbb{R}^N$ as follows

$$\min_{\substack{\hat{\mathbf{y}} \in \mathbb{R}^N, \mathbf{w}_{1j} \in \Delta \\ \|\mathbf{w}_{2j}\|_2 \leq 1, w_{3j} \in \mathbb{R}}} \sum_{i=1}^{N} \mathcal{L}\left(\hat{y}_i, y_i\right) + \beta \|\mathbf{w}_3\|_1 \quad \text{s.t.} \quad \hat{y}_i = \sum_{j=1}^{h} \mathbf{w}_{1j}^\top \mathbf{X}_i \mathbf{w}_{2j} w_{3j}, \; \forall i \in [n]. \quad (18)$$

Now we can form the Lagrangian for (18) as

$$
\begin{aligned}
L(\mathbf{v}, \mathbf{y}, \mathbf{w}_3) &:= \sum_{i=1}^{N} \mathcal{L}\left(\hat{y}_i, y_i\right) + \beta \|\mathbf{w}_3\|_1 + \sum_{i=1}^{N} v_i \left(\hat{y}_i - \sum_{j=1}^{h} \mathbf{w}_{1j}^\top \mathbf{X}_i \mathbf{w}_{2j} w_{3j}\right) \\
&= \sum_{i=1}^{N} \mathcal{L}\left(\hat{y}_i, y_i\right) + \sum_{i=1}^{N} v_i \hat{y}_i + \beta \|\mathbf{w}_3\|_1 - \sum_{i=1}^{N} v_i \sum_{j=1}^{h} \mathbf{w}_{1j}^\top \mathbf{X}_i \mathbf{w}_{2j} w_{3j} \\
&= \sum_{i=1}^{N} \mathcal{L}\left(\hat{y}_i, y_i\right) + \sum_{i=1}^{N} v_i \hat{y}_i + \beta \|\mathbf{w}_3\|_1 - \sum_{j=1}^{h} \sum_{i=1}^{N} v_i \mathbf{w}_{1j}^\top \mathbf{X}_i \mathbf{w}_{2j} w_{3j}
\end{aligned}
$$

Minimizing the Lagrangian $L(\cdot)$ yields the following dual problem of (6)

$$\max_{\mathbf{v} \in \mathbb{R}^N} -\mathcal{L}^*\left(\mathbf{v}, \mathbf{y}\right) \quad \text{s.t.} \quad \max_{\mathbf{w}_1 \in \Delta, \|\mathbf{w}_2\|_2 \leq 1} \left|\sum_{i=1}^{N} v_i \mathbf{w}_1^\top \mathbf{X}_i \mathbf{w}_2\right| \leq \beta, \quad (19)$$

where $\mathcal{L}^* (\cdot)$ denotes the Fenchel congregate function of the original loss function $\mathcal{L} (\cdot)$ (Boyd & Vandenberghe, 2004), which is defined as follows

$$\mathcal{L}^* (\mathbf{v}, \mathbf{y}) := \max_{\mathbf{z} \in \mathbb{R}^N} \mathbf{z}^\top \mathbf{v} - \mathcal{L} (\mathbf{z}, \mathbf{y}).$$

In order to convexify the dual constraint, we next find the maximizers of the dual constraint as follows

$$
\begin{aligned}
\max_{\mathbf{w}_1 \in \Delta, \|\mathbf{w}_2\|_2 \leq 1} \left| \sum_{i=1}^N v_i \mathbf{w}_1^\top \mathbf{X}_i \mathbf{w}_2 \right| &= \max_{\mathbf{w}_1 \in \Delta} \left\| \sum_{i=1}^N v_i \mathbf{w}_1^\top \mathbf{X}_i \right\|_2 \\
&= \max_{\mathbf{w}_1 \in \Delta} \left\| \sum_{i=1}^N \sum_{k=1}^n v_i w_{1k} \mathbf{x}_{ik} \right\|_2 \\
&\leq \max_{\mathbf{w}_1 \in \Delta} \sum_{k=1}^n w_{1k} \left\| \sum_{i=1}^N v_i \mathbf{x}_{ik} \right\|_2 \\
&= \max_{k \in [n]} \left\| \sum_{i=1}^N v_i \mathbf{x}_{ik} \right\|_2,
\end{aligned}
\tag{20}
$$

where the upperbound is achieved when each $\mathbf{w}_1$ has is a vector of zeros except a single one located at the index of maximum norm of weighted tokens.

Based on the observation in (20), we can equivalently write the dual problem in (19) as follows

$$
\begin{aligned}
d^* &= \max_{\mathbf{v} \in \mathbb{R}^N} -\mathcal{L}^* (\mathbf{v}, \mathbf{y}) \quad \text{s.t.} \max_{k \in [h]} \left\| \sum_{i=1}^N v_i \mathbf{x}_{ik} \right\|_2 \leq \beta \\
&= \max_{\mathbf{v} \in \mathbb{R}^N} -\mathcal{L}^* (\mathbf{v}, \mathbf{y}) \quad \text{s.t.} \left\| \sum_{i=1}^N v_i \mathbf{x}_{ik} \right\|_2 \leq \beta, \forall k \in [n].
\end{aligned}
\tag{21}
$$

Next, we form the Lagrangian for the dual problem (21)

$$L(\mathbf{v}, \mathbf{y}, \boldsymbol{\lambda}) := -\mathcal{L}^* (\mathbf{v}, \mathbf{y}) + \sum_{k=1}^n \lambda_k \left( \beta - \left\| \sum_{i=1}^N v_i \mathbf{x}_{ik} \right\|_2 \right)$$

and the corresponding optimization problem can be written in terms of the Lagrangian as

$$\min_{\boldsymbol{\lambda} \geq \mathbf{0}} \max_{\mathbf{v} \in \mathbb{R}^N} L(\mathbf{v}, \mathbf{y}, \boldsymbol{\lambda}) = -\mathcal{L}^* (\mathbf{v}, \mathbf{y}) + \frac{1}{2} \|\mathbf{y}\|_2^2 + \sum_{k=1}^n \lambda_k \left( \beta - \left\| \sum_{i=1}^N v_i \mathbf{x}_{ik} \right\|_2 \right).$$

Then, we introduce additional variables $\mathbf{r}_k \in \mathbb{R}^d$ to equivalently formulate the optimization problem above as

$$\min_{\boldsymbol{\lambda} \geq \mathbf{0}} \max_{\mathbf{v} \in \mathbb{R}^n} \min_{\mathbf{r}_k : \|\mathbf{r}_k\|_2 \leq 1} -\mathcal{L}^* (\mathbf{v}, \mathbf{y}) + \sum_{k=1}^n \lambda_k \left( \beta - \mathbf{r}_k^\top \sum_{i=1}^N v_i \mathbf{x}_{ik} \right).$$

Due to Sion's minimax theorem (Sion, 1958), we can change the order the minimization and maximization to obtain closed-form solutions for the maximization over the dual variable $\mathbf{v}$. This yields the following problem

$$\min_{\boldsymbol{\lambda} \geq \mathbf{0}} \min_{\mathbf{r}_k : \|\mathbf{r}_k\|_2 \leq 1} \sum_{i=1}^N \mathcal{L} \left( \sum_{k=1}^n \lambda_k \mathbf{r}_k^\top \mathbf{x}_{ik}, y_i \right) + \beta \sum_{k=1}^n \lambda_k.$$

Next, we apply a variable change as $\mathbf{z}_k := \lambda_k \mathbf{r}_k$, then the problem above reduces to

$$\min_{\mathbf{z}_k : \|\mathbf{z}_k\|_2 \leq \lambda_k} \sum_{i=1}^N \mathcal{L} \left( \sum_{k=1}^n \mathbf{z}_k^\top \mathbf{x}_{ik}, y_i \right) + \beta \sum_{k=1}^n \lambda_k.$$

From the KKT conditions, we now that $\lambda_k = \|\mathbf{z}_k\|_2$ at a global optimum. In particular, if $\lambda_k > \|\mathbf{z}_k\|_2$, then one can further minimize the objective function by reducing the $\lambda_k$ and therefore $\lambda_k$ would not be optimal. With this, the problem can be reformulated as

$$\min_{\mathbf{z}_k} \sum_{i=1}^{N} \mathcal{L}\left(\sum_{k=1}^{n} \mathbf{z}_k^\top \mathbf{x}_{ik}, y_i\right) + \beta \sum_{k=1}^{n} \|\mathbf{z}_k\|_2,$$

which is the same formulation with (8) and therefore concludes the proof. $\quad\square$

### A.4 STRONG DUALITY PROOF

To get the bidual of (7), we utilize semi-infinite duality theory. We first compute the dual of (21) with respect to the dual parameter $\mathbf{v}$ as follows

$$p_\infty^* := \min_{\boldsymbol{\mu}} \sum_{i=1}^{N} \mathcal{L}\left(\int_{\substack{\mathbf{w}_1 \in \Delta \\ \|\mathbf{w}_2\|_2 \leq 1}} \mathbf{w}_1^\top \mathbf{X}_i \mathbf{w}_2 d\mu(\mathbf{w}_1, \mathbf{w}_2), y_i\right) + \beta\|\boldsymbol{\mu}\|_{TV}, \tag{22}$$

where $\|\boldsymbol{\mu}\|_{TV}$ represents the total variation norm of the signed measure $\boldsymbol{\mu}$. Remark that (22) is an infinite-dimensional dimensional training problem such as the ones in Bach (2017). Also, notice that this problem is convex with respect to the linear measure $\mu$ (Bach, 2017). Therefore, strong duality holds, i.e., $d^* = p_\infty^*$ where $d^*$ denotes the objective value of (21). In addition to this, although (22) is an infinite-dimensional problem, it has at most $N + 1$ heads at the optimum due to Caratheodory's theorem (Rosset et al., 2007). Therefore, (22) is equivalent to the following problem

$$p_\infty^* = \min_{\substack{\mathbf{w}_{1j} \in \Delta \\ \|\mathbf{w}_{2j}\|_2 \leq 1}} \sum_{i=1}^{N} \mathcal{L}\left(\sum_{j=1}^{h^*} \mathbf{w}_{1j}^\top \mathbf{X}_i \mathbf{w}_{2j} w_{3j}, y_i\right) + \beta\|\mathbf{w}_3\|_1 \tag{23}$$

where $h^* \leq N + 1$. We note that that provided that $h \geq h^*$, (23) and (7) are the same problems, which proves strong duality, i.e., $p^* = p_\infty^* = d^*$, where $p^*$ denotes the objective value of (7). $\quad\square$

### A.5 PROOF OF THEOREM 2

We first apply the scaling technique in Lemma 1 for $\{\mathbf{w}_{2j}, \mathbf{w}_{3j}\}_{j=1}^{m}$

$$\bar{\mathbf{w}}_{2j} := \alpha_j \mathbf{w}_{2j}, \ \bar{\mathbf{w}}_{3j} := \frac{\mathbf{w}_{3j}}{\alpha_j}.$$

Then, following the same steps in Lemma 1, (9) can be equivalently formulated as

$$\min_{\substack{\mathbf{w}_{1j} \in \Delta \\ \mathbf{w}_{2j} : \|\mathbf{w}_{2j}\|_2 \leq 1 \\ \mathbf{w}_{3j} \in \mathbb{R}^c}} \sum_{i=1}^{N} \mathcal{L}\left(\sum_{j=1}^{h} \mathbf{w}_{1j}^\top \mathbf{X}_i \mathbf{w}_{2j} \mathbf{w}_{3j}, \mathbf{y}_i\right) + \frac{\beta}{2} \sum_{j=1}^{h} \|\mathbf{w}_{3j}\|_1. \tag{24}$$

Next, we again construct the Lagrangian function by introducing an additional variable $\hat{\mathbf{y}}_i \in \mathbb{R}^c, \forall i \in [N]$ as follows

$$\min_{\substack{\hat{\mathbf{y}}_i \in \mathbb{R}^c, \mathbf{w}_{1j} \in \Delta \\ \|\mathbf{w}_{2j}\|_2 \leq 1, w_{3j} \in \mathbb{R}}} \sum_{i=1}^{N} \mathcal{L}(\hat{\mathbf{y}}_i, \mathbf{y}_i) + \beta \sum_{j=1}^{h} \|\mathbf{w}_{3j}\|_1 \quad \text{s.t.} \quad \hat{\mathbf{y}}_i = \sum_{j=1}^{h} \mathbf{w}_{1j}^\top \mathbf{X}_i \mathbf{w}_{2j} \mathbf{w}_{3j}, \ \forall i \in [N]. \tag{25}$$

Now we can form the Lagrangian for (25) as

$$L\left(\{\mathbf{v}_i\}_{i=1}^{N}, \mathbf{y}, \{\mathbf{w}_{3j}\}_{j=1}^{h}\right) := \sum_{i=1}^{N} \mathcal{L}(\hat{\mathbf{y}}_i, \mathbf{y}_i) + \beta \sum_{j=1}^{h} \|\mathbf{w}_{3j}\|_1 + \sum_{i=1}^{N} \mathbf{v}_i^\top \left(\hat{\mathbf{y}}_i - \sum_{j=1}^{h} \mathbf{w}_{1j}^\top \mathbf{X}_i \mathbf{w}_{2j} \mathbf{w}_{3j}\right)$$

$$= \sum_{i=1}^{N} \mathcal{L}(\hat{\mathbf{y}}_i, \mathbf{y}_i) + \sum_{i=1}^{N} \mathbf{v}_i^\top \hat{\mathbf{y}}_i + \beta \sum_{j=1}^{h} \|\mathbf{w}_{3j}\|_1 - \sum_{i=1}^{N} \mathbf{v}_i^\top \sum_{j=1}^{h} \mathbf{w}_{1j}^\top \mathbf{X}_i \mathbf{w}_{2j} \mathbf{w}_{3j}$$

$$= \sum_{i=1}^{N} \mathcal{L}(\hat{\mathbf{y}}_i, \mathbf{y}_i) + \sum_{i=1}^{N} \mathbf{v}_i^\top \hat{\mathbf{y}}_i + \beta \sum_{j=1}^{h} \|\mathbf{w}_{3j}\|_1 - \sum_{j=1}^{h} \sum_{i=1}^{N} \mathbf{w}_{1j}^\top \mathbf{X}_i \mathbf{w}_{2j} \mathbf{v}_i^\top \mathbf{w}_{3j}$$

Minimizing the Lagrangian $L(\cdot)$ yields the following dual problem of (9)

$$\max_{\mathbf{v}_i \in \mathbb{R}^c} -\mathcal{L}^* \left( \{\mathbf{v}_i\}_{i=1}^N, \{\mathbf{y}_i\}_{i=1}^N \right) \quad \text{s.t.} \quad \max_{\mathbf{w}_1 \in \Delta, \|\mathbf{w}_2\|_2 \leq 1} \left\| \sum_{i=1}^N \mathbf{v}_i \mathbf{w}_1^\top \mathbf{X}_i \mathbf{w}_2 \right\|_\infty \leq \beta, \qquad (26)$$

where $\mathcal{L}^* (\cdot)$ denotes the Fenchel congregate function of the original loss function $\mathcal{L} (\cdot)$ (Boyd & Vandenberghe, 2004), which is defined as follows

$$\mathcal{L}^* \left( \{\mathbf{v}_i\}_{i=1}^N, \{\mathbf{y}_i\}_{i=1}^N \right) := \max_{\mathbf{Z} \in \mathbb{R}^{N \times c}} \operatorname{trace} \left( \mathbf{Z}^\top \mathbf{V} \right) - \mathcal{L} \left( \mathbf{Z}, \mathbf{Y} \right),$$

where $\mathbf{V}, \mathbf{Y} \in \mathbb{R}^{N \times c}$ are the matrix representations for the set of variables $\{\mathbf{v}_i, \mathbf{y}_i\}_{i=1}^N$. In order to characterize the optimal layer weight explicitly, we next find the maximizers of the dual constraint as follows

$$\max_{\mathbf{w}_1 \in \Delta, \|\mathbf{w}_2\|_2 \leq 1} \left\| \sum_{i=1}^N \mathbf{v}_i \mathbf{w}_1^\top \mathbf{X}_i \mathbf{w}_2 \right\|_\infty = \max_{\mathbf{w}_1 \in \Delta} \max_{l \in [c]} \left\| \sum_{i=1}^N v_{il} \mathbf{w}_1^\top \mathbf{X}_i \right\|_2$$

$$= \max_{\mathbf{w}_1 \in \Delta} \max_{l \in [c]} \left\| \sum_{i=1}^N \sum_{k=1}^n v_{il} w_{1k} \mathbf{x}_{ik} \right\|_2$$

$$\leq \max_{\mathbf{w}_1 \in \Delta} \max_{l \in [c]} \sum_{k=1}^n w_{1k} \left\| \sum_{i=1}^N v_i \mathbf{x}_{ik} \right\|_2$$

$$= \max_{l \in [c]} \max_{k \in [n]} \left\| \sum_{i=1}^N v_{il} \mathbf{x}_{ik} \right\|_2, \qquad (27)$$

where the upperbound is achieved when each $\mathbf{w}_1$ has is a vector of zeros except a single one located at the index of maximum norm of weighted tokens.

Based on the observation in (27), we can equivalently write the dual problem in (26) as follows

$$\max_{\mathbf{v} \in \mathbb{R}^N} -\mathcal{L}^* \left( \{\mathbf{v}_i\}_{i=1}^N, \{\mathbf{y}_i\}_{i=1}^N \right) \quad \text{s.t.} \quad \left\| \sum_{i=1}^N v_{il} \mathbf{x}_{ik} \right\|_2 \leq \beta, \forall k \in [n], \forall l \in [c]. \qquad (28)$$

Then directly following the steps in the proof of Theorem 1 yields the following convex optimization problem

$$\min_{\mathbf{Z}_l \in \mathbb{R}^{n \times d}} \sum_{i=1}^N \sum_{l=1}^c \mathcal{L} \left( \operatorname{trace} \left( \mathbf{Z}_l^\top \mathbf{X}_i \right), y_{il} \right) + \beta \sum_{l=1}^c \sum_{k=1}^n \|\mathbf{z}_{lk}\|_2.$$

$\square$

## A.6 PROOF OF THEOREM 3

We first apply the scaling technique in Lemma 1 for $\{\mathbf{w}_{2j}, \mathbf{w}_{3j}\}_{j=1}^m$

$$\bar{\mathbf{w}}_{2j} := \alpha_j \mathbf{w}_{2j}, \ \bar{\mathbf{w}}_{3j} := \frac{\mathbf{w}_{3j}}{\alpha_j}.$$

Then, following the same steps in Lemma 1, (11) can be equivalently formulated as

$$\min_{\substack{\mathbf{w}_{1j} \in \Delta \\ \mathbf{w}_{2j}: \|\mathbf{w}_{2j}\|_2 \leq 1 \\ \mathbf{w}_{3j} \in \mathbb{R}^c}} \sum_{i=1}^N \mathcal{L} \left( \sum_{j=1}^h \sigma \left( \mathbf{w}_{1j}^\top \mathbf{X}_i \mathbf{w}_{2j} \right) \mathbf{w}_{3j}, \mathbf{y}_i \right) + \frac{\beta}{2} \sum_{j=1}^h \|\mathbf{w}_{3j}\|_1. \qquad (29)$$

Next, we again construct the Lagrangian function by introducing an additional variable $\hat{\mathbf{y}}_i \in \mathbb{R}^c, \forall i \in [N]$ as follows

$$\min_{\substack{\hat{\mathbf{y}}_i \in \mathbb{R}^c, \mathbf{w}_{1j} \in \Delta \\ \|\mathbf{w}_{2j}\|_2 \leq 1, w_{3j} \in \mathbb{R}}} \sum_{i=1}^N \mathcal{L} \left( \hat{\mathbf{y}}_i, \mathbf{y}_i \right) + \beta \sum_{j=1}^h \|\mathbf{w}_{3j}\|_1 \quad \text{s.t.} \quad \hat{\mathbf{y}}_i = \sum_{j=1}^h \sigma \left( \mathbf{w}_{1j}^\top \mathbf{X}_i \mathbf{w}_{2j} \right) \mathbf{w}_{3j}, \ \forall i \in [N].$$

$$(30)$$

Now we can form the Lagrangian for (30) as

$$
L\left(\{\mathbf{v}_i\}_{i=1}^N, \mathbf{y}, \{\mathbf{w}_{3j}\}_{j=1}^h\right) := \sum_{i=1}^N \mathcal{L}\left(\hat{\mathbf{y}}_i, \mathbf{y}_i\right) + \beta \sum_{j=1}^h \|\mathbf{w}_{3j}\|_1 + \sum_{i=1}^N \mathbf{v}_i^\top \left(\hat{\mathbf{y}}_i - \sum_{j=1}^h \sigma\left(\mathbf{w}_{1j}^\top \mathbf{X}_i \mathbf{w}_{2j}\right) \mathbf{w}_{3j}\right)
$$

$$
= \sum_{i=1}^N \mathcal{L}\left(\hat{\mathbf{y}}_i, \mathbf{y}_i\right) + \sum_{i=1}^N \mathbf{v}_i^\top \hat{\mathbf{y}}_i + \beta \sum_{j=1}^h \|\mathbf{w}_{3j}\|_1 - \sum_{i=1}^N \mathbf{v}_i^\top \sum_{j=1}^h \sigma\left(\mathbf{w}_{1j}^\top \mathbf{X}_i \mathbf{w}_{2j}\right) \mathbf{w}_{3j}
$$

$$
= \sum_{i=1}^N \mathcal{L}\left(\hat{\mathbf{y}}_i, \mathbf{y}_i\right) + \sum_{i=1}^N \mathbf{v}_i^\top \hat{\mathbf{y}}_i + \beta \sum_{j=1}^h \|\mathbf{w}_{3j}\|_1 - \sum_{j=1}^h \sum_{i=1}^N \sigma\left(\mathbf{w}_{1j}^\top \mathbf{X}_i \mathbf{w}_{2j}\right) \mathbf{v}_i^\top \mathbf{w}_{3j}
$$

Minimizing the Lagrangian $L(\cdot)$ yields the following dual problem of (11)

$$
\max_{\mathbf{v}_i \in \mathbb{R}^c} -\mathcal{L}^*\left(\{\mathbf{v}_i\}_{i=1}^N, \{\mathbf{y}_i\}_{i=1}^N\right) \quad \text{s.t.} \max_{\mathbf{w}_{1j} \in \Delta, \|\mathbf{w}_{2j}\|_2 \le 1} \left\|\sum_{i=1}^N \mathbf{v}_i \sigma\left(\mathbf{w}_{1j}^\top \mathbf{X}_i \mathbf{w}_{2j}\right)\right\|_\infty \le \beta, \forall j \in [h],
\tag{31}
$$

where $\mathcal{L}^*(\cdot)$ denotes the Fenchel congregate function of the original loss function $\mathcal{L}(\cdot)$ (Boyd & Vandenberghe, 2004), which is defined as follows

$$
\mathcal{L}^*\left(\{\mathbf{v}_i\}_{i=1}^N, \{\mathbf{y}_i\}_{i=1}^N\right) := \max_{\mathbf{Z} \in \mathbb{R}^{N \times c}} \operatorname{trace}\left(\mathbf{Z}^\top \mathbf{V}\right) - \mathcal{L}\left(\mathbf{Z}, \mathbf{Y}\right),
$$

where $\mathbf{V}, \mathbf{Y} \in \mathbb{R}^{N \times c}$ are the matrix representations for the set of variables $\{\mathbf{v}_i, \mathbf{y}_i\}_{i=1}^N$.

We next note that we utilize the gated ReLU nonlinearity introduced in Mishkin et al. (2022). Thus, the activations $\sigma\left(\mathbf{w}_{1j}^\top \mathbf{X}_i \mathbf{w}_{2j}\right)$ can be expressed as

$$
\sigma\left(\mathbf{w}_{1j}^\top \mathbf{X}_i \mathbf{w}_{2j}\right) := \mathbb{1}_{ij} \mathbf{w}_{1j}^\top \mathbf{X}_i \mathbf{w}_{2j},
$$

where $\mathbb{1}_{ij} := \mathbb{1}\left\{\mathbf{u}_{1j}^\top \mathbf{X}_i \mathbf{u}_{2j} \ge 0\right\}$ and here $\{\mathbf{u}_{1j}, \mathbf{u}_{2j}\}_{j=1}^h$ are fixed vectors that can be randomly selected. For instance, a common choice is $\mathbf{u}_{1j} \sim \mathcal{N}(\mathbf{0}, \mathbf{I}_n)$ and $\mathbf{u}_{2j} \sim \mathcal{N}(\mathbf{0}, \mathbf{I}_d)$. For the rest of the derivations, we use this equivalent formulation of the activation function.

In order to characterize the optimal layer weight explicitly, we next find the maximizers of the dual constraint as follows

$$
\max_{j \in [h]} \max_{\mathbf{w}_{1j} \in \Delta, \|\mathbf{w}_{2j}\|_2 \le 1} \left\|\sum_{i=1}^N \mathbf{v}_i \sigma\left(\mathbf{w}_{1j}^\top \mathbf{X}_i \mathbf{w}_{2j}\right)\right\|_\infty = \max_{j \in [h]} \max_{l \in [c]} \max_{\mathbf{w}_{1j} \in \Delta} \left\|\sum_{i=1}^N v_{il} \mathbb{1}_{ij} \mathbf{w}_{1j}^\top \mathbf{X}_i\right\|_2
$$

$$
= \max_{j \in [h]} \max_{\mathbf{w}_{1j} \in \Delta} \max_{l \in [c]} \left\|\sum_{i=1}^N \sum_{k=1}^n v_{il} \mathbb{1}_{ij} w_{1jk} \mathbf{x}_{ik}\right\|_2
$$

$$
\le \max_{j \in [h]} \max_{\mathbf{w}_{1j} \in \Delta} \max_{l \in [c]} \sum_{k=1}^n w_{1jk} \left\|\sum_{i=1}^N v_i \mathbb{1}_{ij} \mathbf{x}_{ik}\right\|_2
$$

$$
= \max_{j \in [h]} \max_{l \in [c]} \max_{k \in [n]} \left\|\sum_{i=1}^N v_{il} \mathbb{1}_{ij} \mathbf{x}_{ik}\right\|_2,
\tag{32}
$$

where the upperbound is achieved when each $\mathbf{w}_{1j}$ has is a vector of zeros except a single one located at the index of maximum norm of weighted tokens.

Based on the observation in (32), we can equivalently write the dual problem in (31) as follows

$$
\max_{\mathbf{v} \in \mathbb{R}^N} -\mathcal{L}^*\left(\{\mathbf{v}_i\}_{i=1}^N, \{\mathbf{y}_i\}_{i=1}^N\right) \quad \text{s.t.} \left\|\sum_{i=1}^N v_{il} \mathbb{1}_{ij} \mathbf{x}_{ik}\right\|_2 \le \beta, \forall k \in [n], \forall l \in [c], \forall j \in [h].
\tag{33}
$$

Then directly following the steps in the proof of Theorem 2 yields the following convex optimization problem

$$
\min_{\mathbf{Z}_{jl} \in \mathbb{R}^{n \times d}} \sum_{i=1}^N \sum_{l=1}^c \mathcal{L}\left(\sum_{j=1}^h \mathbb{1}_{ij} \operatorname{trace}\left(\mathbf{Z}_{jl}^\top \mathbf{X}_i\right), y_{il}\right) + \beta \sum_{l=1}^c \sum_{j=1}^h \sum_{k=1}^n \|\mathbf{z}_{jlk}\|_2.
$$

$\square$

### A.7 MATRIX TARGETS

We now consider the following vector output attention based model training problem, where the targets are vectors, i.e., $\mathbf{Y}_i \in \mathbb{R}^{n \times c}$,

$$\min_{\substack{\mathbf{W}_{1j} \in \Delta \\ \mathbf{W}_{2j} \in \mathbb{R}^{d \times c}, w_{3j} \in \mathbb{R}}} \sum_{i=1}^{N} \mathcal{L}\left(\sum_{j=1}^{h} \mathbf{W}_{1j} \mathbf{X}_i \mathbf{W}_{2j} w_{3j}, \mathbf{Y}_i\right) + \frac{\beta}{2} \sum_{j=1}^{h} \left(\|\mathbf{W}_{2j}\|_1^2 + (w_{3j})^2\right). \quad (34)$$

Then the corresponding dual problem is as follows

$$\max_{\mathbf{V}_i \in \mathbb{R}^{n \times c}} -\sum_{i=1}^{N} \mathcal{L}^*\left(\mathbf{V}_i, \mathbf{Y}_i\right) \quad \text{s.t.} \quad \max_{\mathbf{W}_1 \in \Delta, \|\mathbf{W}_2\|_2 \leq 1} \left|\text{trace}\left(\sum_{i=1}^{N} \mathbf{V}_i^\top \mathbf{W}_1 \mathbf{X}_i \mathbf{W}_2\right)\right| \leq \beta. \quad (35)$$

In order to characterize the optimal layer weight explicitly, we next find the maximizers of the dual constraint as follows

$$\max_{\mathbf{W}_1 \in \Delta, \|\mathbf{W}_2\|_2 \leq 1} \left|\text{trace}\left(\sum_{i=1}^{N} \mathbf{V}_i^\top \mathbf{W}_1 \mathbf{X}_i \mathbf{W}_2\right)\right| = \max_{\mathbf{W}_1 \in \Delta} \left\|\sum_{i=1}^{N} \mathbf{v}_i^\top \mathbf{W}_1 \mathbf{X}_i\right\|_\infty$$

$$= \max_{l \in [c]} \max_{j \in [d]} \max_{\mathbf{W}_1 \in \Delta} \left|\sum_{i=1}^{N} \mathbf{v}_{il}^\top \mathbf{W}_1 \mathbf{x}_{ij}\right| \quad (36)$$

Based on the equivalent formulation in (36), the dual problem in (35) can be equivalently written as

$$\max_{\mathbf{V}_i \in \mathbb{R}^{n \times c}} -\sum_{i=1}^{N} \mathcal{L}^*\left(\mathbf{V}_i, \mathbf{Y}_i\right) \quad \text{s.t.} \quad \max_{\mathbf{W}_1 \in \Delta} \left|\sum_{i=1}^{N} \mathbf{v}_{il}^\top \mathbf{W}_1 \mathbf{x}_{ij}\right| \leq \beta, \ \forall j \in [d], \ \forall l \in [c].$$

The rest of the derivations directly follows from the proof of Theorem 2 and yields the following result.

**Theorem A.1.** *Based on the characterization of the dual constraint in (36), the non-convex optimization problem (34) can be equivalently cast as the following convex optimization problem*

$$\min_{\mathbf{Z}_{jl}^{(1)}, \mathbf{Z}_{jl}^{(2)} \in \mathbb{R}_+^{n \times n}} \sum_{i=1}^{N} \sum_{l=1}^{c} \mathcal{L}\left(\sum_{j=1}^{d} \left(\mathbf{Z}_{jl}^{(1)} - \mathbf{Z}_{jl}^{(2)}\right) \mathbf{x}_{ij}, \mathbf{y}_{il}\right) 2^2 + \beta \sum_{l=1}^{c} \sum_{j=1}^{d} \left(\left\|\mathbf{Z}_{jl}^{(1)}\right\|_{1,\infty} + \left\|\mathbf{Z}_{jl}^{(2)}\right\|_{1,\infty}\right). \quad (37)$$

**Remark A.2.** *Instead of the non-convex formulation in (34), we can also start from the following formulation in this setting*

$$\min_{\substack{\mathbf{W}_{1j} \in \Delta \\ \mathbf{w}_{2j} \in \mathbb{R}^d, \mathbf{w}_{3j} \in \mathbb{R}^c}} \sum_{i=1}^{N} \mathcal{L}\left(\sum_{j=1}^{h} \mathbf{W}_{1j} \mathbf{X}_i \mathbf{w}_{2j} \mathbf{w}_{3j}^\top, \mathbf{Y}_i\right) F^2 + \frac{\beta}{2} \sum_{j=1}^{h} \left(\|\mathbf{w}_{2j}\|_1^2 + \|\mathbf{w}_{3j}\|_1^2\right),$$

*which also yields the convex formulation in (37).*

