# OpenReview forum: "Convexifying Transformers: Improving optimization and understanding of transformer networks"
_ICLR.cc/2024/Conference — Submitted to ICLR 2024_

### Official Review · Reviewer_E73C · 2023-10-29

**Soundness:** 2 fair
**Presentation:** 3 good
**Contribution:** 2 fair
**Rating:** 5
**Confidence:** 3

**Summary:**

This paper proposes a convex alternative to the self-attention mechanism in transformers. which reveals an implicit regularization mechanism.

**Strengths:**

1.	The paper is well-written and logical.
2.	The paper is interesting in the idea of making the attention mechanism convex and easy to optimize. The discovery of the implicit regularization mechanism in the convex optimization problem revealed in the paper is of significant inspiration.

**Weaknesses:**

1.	The use of "various" in the experimental description when stating the contributions is inaccurate. The presentation of the paper is clear, but I am concerned about the experimental section. I believe the authors should conduct experiments on a wider range of benchmarks to validate the proposed alternative convex attention, particularly in comparison with the multi-layer perceptron and matrix decomposition architectures.
2.	Since this paper proposes an alternative to self-attention, experiments should be carried out on more downstream tasks to validate.

**Questions:**

see above

---

### Official Review · Reviewer_6FNQ · 2023-10-31

**Soundness:** 2 fair
**Presentation:** 3 good
**Contribution:** 2 fair
**Rating:** 5
**Confidence:** 4

**Summary:**

In this paper, the authors proposes a novel convex analytic approach to improve the understanding and optimization of transformers. More specifically, they introduce a convex alternative to substitute the nonconvex softmax operation typically used in transformer networks and remove all nonconvex heuristics such as skip connection, layer norm. This alternative approach simplifies the problem into a convex optimization problem. They conducted a series of experiments, comparing the traditional nonconvex transformer with their proposed convex version. The results demonstrate that the convex version achieves faster convergence and lower test loss, highlighting the advantages of their approach.

**Strengths:**

1. Simplifying the transformer into a convex problem streamlines theoretical analysis.
2. The convex alternative shows good convergence speed on the demonstrated tasks, and the reformulation can mitigate the grokking phenomenon
3. The sparsity-inducing regularization which aims to use as few tokens as possible to fit the training labels seems to be a interesting finding.

**Weaknesses:**

1. The oversimplification of the problem is a concern. Removing the softmax component, although it may simplify analyses, fails to provide genuine insights into the actual functioning of the transformer. It does little to enhance our understanding of the transformer architecture, which heavily relies on the softmax operation.
2. The absence of a theoretical proof regarding the convergence rate of the proposed alternative version is another limitation of this work.
3. The chosen tasks for demonstration may be too easy, potentially limiting the generalizability and applicability of the findings.

**Questions:**

1. It would be better to have a theoretically convergence rate analysis on the reformulation.
2. Can the proposed convex alternative be scaled up to a larger size (e.g., billions of parameters).
3. How does this approach compare to other transformer alternatives that have also eliminated the softmax operation, such as Skyformer [a][b]?

[a] Skyformer: Remodel Self-Attention with Gaussian Kernel and Nyström Method, Chen et al., NeurIPS 2021
[b] Soft: Softmax-free transformer with linear complexity, Lu et al. NeurIPS

---

### Official Review · Reviewer_boLQ · 2023-11-01

**Soundness:** 2 fair
**Presentation:** 3 good
**Contribution:** 3 good
**Rating:** 5
**Confidence:** 3

**Summary:**

This work presents a way to convexify transformers and improve their understanding. The authors first introduce a convex alternative to the self-attention mechanism. Then they consider a regularized training objective for a transformer model and proceed to cast it into a convex optimization problem. They also show that this reveals an implicit regularization mechanism that encourages sparsity across tokens. In addition, they include experimental results on some small problem instances to show the improvement as a result of convexifying the model and the training objective. They first consider student teacher setting with pretrained BERT and then, they consider the algorithmic dataset to show that proposed alternative mitigates grokking.

**Strengths:**

1. The proposed convex alternative to self-attention presents a natural way to interpret the mechanism. It is interesting to see that an implicit regularization on token sparsity shows up after convexifying the regularized training objective.

2. The paper is generally nice to read and follow.

**Weaknesses:**

1. ****Some statements don't accurately reflect the contributions, and should be rephrased/clarified.****

- Equation (4) seems closer to a version of (3) without the weight decay terms for $W_q$ and $W_k$. But that is a specific type of regularization, which should be clarified in the introduction/contributions. I also suggest including an experiment on this to see how it compares with (3) and (4).
- The statement that equation (3) can be reformulated as eq. (4) is not accurate. These two are not equivalent.
    - Although there exists a $W_1$ for every value of the softmax($XW_QW_k^TX^T$), the inverse may not be true.
    - The weight decay term for $W_q$ and $W_k$ does not necessarily translate into the simplex constraint on $W_1$.

    Some discussion on this should be added.

- Some claims are too strong: e.g. “we apply our framework to various architectures” seems to suggest that the authors also consider skip connections or layer normalization components of the conventional attention block as well; Fig. 1 also suggests this, while this work does not considers these components. This should be clarified.

2. ****The experimental results seem limited, and the claim that the proposed alternative is effective is not well-supported.****

- The experimental results in Figs. 2 and 3 are not very convincing. Figs. 2 and 3 shows that (10) shows improvement over both (3) and (9). However, it is not very clear whether the improvement in (10) is solely due to convexification, and not a result of other factors, such as hyperparameter values or the type and strength of the regularization term in (3) and (9). It would be good to consider other objectives for comparisons, such as (5), (3) without the weight decay terms for $W_q$ and $W_k$ as well as (3) without weight decay, and so on, for a better picture of the effect of the proposed alternative as well as how similarly do the various objectives considered in the paper behave.
- In addition, Figs. 2 and 3 only consider the attention only model. Some exprimental results for comparing the three models depicted in Fig. 1 should be included to support the claims better.
- Overall, the claim that the approach is effective would be better supported by including a more thorough evaluation.

**Questions:**

1. For experiments:

- Details about hyperparameters and range for the grid search seem missing and would be helpful to add these.

- How are the train and test error for the BERT experiment calculated for the convex problem?

2. Some typos, e.g. Table 1 states that n is # of tokens and h is number of heads, while Section 2 states that the data sample has h tokens.

---

### Official Review · Reviewer_Mygf · 2023-11-01

**Soundness:** 3 good
**Presentation:** 4 excellent
**Contribution:** 3 good
**Rating:** 5
**Confidence:** 4

**Summary:**

This paper reformulates the training problem of the attention layer as a convex optimization problem. Then, the authors introduce a novel convex self-attention based on this formulation. Implicit regularization such that sparsity across tokens of self-attention can then be explained by their derivation from the convex alternative. They develop the theory for multi-head attention followed by an additional Fully Connected layer and gated ReLU activation with some relaxations. The first experiment is fine-tuning BERT with student-teacher settings and some previous layers frozen. Then, the last attention layer is then replaced by convex attention to achieve faster convergence and better test error. The second experiment is performed on the task of calculating modular division operation to demonstrate that the convex model improves the convergence speed and training loss for one and two-layer transformer networks.

**Strengths:**

1. The paper is well-written and easy to read.

2. Theoretical derivations are easy to follow.

**Weaknesses:**

1. In session 3.1, the relaxation of $\text{softmax} X W_q W_k X$ to be an arbitrary linear operation $W$ with unit simplex constraints is not realistic and is hard to achieve with vanilla softmax attention. Hence, it is not a good assumption used for analyzing properties of self-attention.

2. Whether the group sparsity across the token indices recovered by the relaxed model in theoretical analysis holds true for vanilla softmax attention is not clear. It will be beneficial to add an experiment to support this point.

3. Except for the BERT experiment, there is not any large-scale task to demonstrate the performance of the proposed attention. This BERT experiment itself is limited since the convex attention layer is not trained with other layers but only trained with the fixed input from the pre-trained layers. It is not clear whether the good performance observed in the paper still holds when training from scratch.

4. The experiment with algorithmic datasets is inadequate to verify whether the proposed attention can yield better accuracy than the vanilla transformer when stacked on each other in my opinion. It will be beneficial to include more experiments trained on other practical tasks such as image classification on ImageNet.

**Questions:**

1. In Figure 3, how is the attention map for the nonconvex training of the BERT model obtained? It is strange to observe that the standard model learns a uniform attention map.

---

### Meta-Review · Area_Chair_RM17 · 2023-12-08

**Metareview:**

This paper addresses to improve transformer models by proposing a convex self-attention layer. The paper is well written and follows an interesting idea. It received borderline reject reviews raining several questions regarding required clarifications and improvements on experiments. Unfortunately, no rebuttal was provided. The paper is not ready for acceptance in its current form.

**Justification For Why Not Higher Score:**

While the paper seems overall of good quality, there are several open questions. No rebuttal was provided.

**Justification For Why Not Lower Score:**

N/A

---

### Decision · Program_Chairs · 2024-01-16

Reject